# *Clostridium butyricum* Can Promote Bone Development by Regulating Lymphocyte Function in Layer Pullets

**DOI:** 10.3390/ijms24021457

**Published:** 2023-01-11

**Authors:** Mengze Song, Xuesong Zhang, Guijuan Hao, Hai Lin, Shuhong Sun

**Affiliations:** 1Department of Preventive Veterinary Medicine, College of Veterinary Medicine, Shandong Agricultural University, Tai’an 271018, China; 2Shandong Provincial Key Laboratory of Animal Biotechnology and Disease Control and Prevention, Shandong Agricultural University, Tai’an 271018, China

**Keywords:** *Clostridium butyricum*, bone, immune system, lymphocytes, Treg, osteoblasts

## Abstract

Bone health problems are a serious threat to laying hens; microbiome-based therapies, which are harmless and inexpensive, may be an effective solution for bone health problems. Here, we examined the impacts of supplementation with *Clostridium butyricum* (CB) on bone and immune homeostasis in pullets. The results of in vivo experiments showed that feeding the pullets CB was beneficial to the development of the tibia and upregulated the levels of the bone formation marker alkaline phosphatase and the marker gene *runt*-*related transcription factor 2 (RUNX2)*. For the immune system, CB treatment significantly upregulated *IL-10* expression and significantly increased the proportion of T regulatory (Treg) cells in the spleen and peripheral blood lymphocytes. In the in vitro test, adding CB culture supernatant or butyrate to the osteoblast culture system showed no significant effects on osteoblast bone formation, while adding lymphocyte culture supernatant significantly promoted bone formation. In addition, culture supernatants supplemented with treated lymphocytes (pretreated with CB culture supernatants) stimulated higher levels of bone formation. In sum, the addition of CB improved bone health by modulating cytokine expression and the ratio of Treg cells in the immune systems of layer pullets. Additionally, in vitro CB could promote the bone formation of laying hen osteoblasts through the mediation of lymphocytes.

## 1. Introduction

Bone is a complex, living, and constantly changing tissue. The calcium that laying hens obtain from food is rapidly deposited in the bones and then released from them during egg laying to contribute to eggshell production [1]. This repetitive process leads to bone calcium loss, which eventually leads to osteoporosis and fractures, causing serious animal welfare problems and economic losses [2]. New research suggests that microbiome-based therapies are harmless and could be an inexpensive and effective way to address bone health problems [3]. The gut is a complex ecosystem made up of trillions of symbiotic bacteria, fungi, Archaea, and viruses; this microbiota is important for maintaining normal gut development and physiological function, as it affects nutrient digestion, metabolism, tissue development, and immune function [4,5]. The metabolites of gut microbes can directly regulate bone metabolism. For example, short-chain fatty acids (SCFAs) produced by the gut microbiota metabolism can affect the differentiation and function of osteoblasts [6,7] and osteoclasts [8]. The immune system and the bone system are closely interconnected, and gut microbiota can indirectly affect bone health by modulating the immune system [9]. Bacteria-derived SCFAs are well-known regulators of immune cells that promote the induction and activity of T regulatory (Treg) cells, thereby inhibiting immune cell responses [10]. Treg cells have the ability to regulate bone tissue, on the one hand blunting bone resorption [11,12] and on the other hand stimulating bone formation by promoting the differentiation of osteoblasts [13].

Scientific research has demonstrated the positive effects of probiotics on bone health in humans and in various animals, including poultry [14]. *Clostridium butyricum* (CB) is an anaerobic butyrate-producing bacterium that is found in a wide variety of environments, including soil, farmed dairy products, and vegetables. CB modulates the composition of the gut microbiota, increasing beneficial bacterial taxa, such as *Lactobacilli* and *Bifidobacteria*, which can ferment undigested dietary fiber and produce SCFAs (mainly butyrate and acetate) in the gut [15]. Numerous studies have shown that CB improves poultry performance [16,17,18,19,20,21,22]. For instance, adding 1 × 10^8^ CFU/kg of CB to the feed was found to improve the egg quality of yellow-feathered broilers [16], and 1 × 10^9^ CFU/kg of CB was found to improve the growth performance of Arbor Acres broilers, promote immune and intestinal barrier function, and benefit cecal flora [17]. Moreover, 2.5 × 10^9^ CFU/kg of CB could improve gut microbiota homeostasis in male Ross 308 broilers [18], and 5 × 10^8^ CFU/kg and 1 × 10^9^ CFU/kg of CB were found to be beneficial for the growth performance, antioxidant activities, and immune function of Arbor Acres broilers [19]. Other research showed that 1 × 10^9^ CFU/kg of CB could improve growth performance, immune response, antioxidant activities, bone development, and gut microbiota in Cobb broilers [20] and that 2 × 10^7^ CFU/kg and 3 × 10^7^ CFU/kg of CB could enhance the growth performance and immune function of male Lingnan yellow broilers and was beneficial to the balance of intestinal flora [21]. Finally, 5 × 10^7^ CFU/kg of CB could increase egg production and eggshell strength in 48 week old Jinghong No. 1 laying hens, as well as promote intestinal immune development and intestinal flora homeostasis [22]. Previous studies by our group have also shown that CB can regulate the intestinal flora, immune function, and resistance to *Salmonella* enteritidis in poultry [23,24,25].

A large number of studies have shown that CB plays a significant role in improving the intestinal barrier, regulating the intestinal flora, and improving the immune function of broilers [16,17,18,19,20,21,22]. However, few studies have examined the effect of CB on laying hens, and research on the bones of laying hens in particular is scant. Given the regulatory effect of CB on poultry immunity and the close connection between the immune system and bones, we conducted a related study on the effects of CB on the bones and the immunity of laying hens. We explored the effects of CB and sodium butyrate on the bone development and production performance of laying hens in different periods in our previous experiments. In one experiment, 1% sodium butyrate and CB at three gradients of 1 × 10^7^ CFU/kg, 1 × 10^8^ CFU/kg, and 1 × 10^9^ CFU/kg were added to the feed. Among these gradients, 1 × 10^8^ CFU/kg and 1 × 10^9^ CFU/kg of CB could significantly increase the tibia index of Hailan brown layer hens in the brooding stage (unpublished data). CB increased mean egg weight and eggshell weight in early and peak laying hens but had no significant effect on bone development (unpublished data). Here, we examined the role of CB in the regulation of bone homeostasis in layer pullets. We found that CB supplementation increased systemic Treg numbers and stimulated bone formation. Examination of the role of CB in bone formation revealed a pathway whereby interaction between CB and lymphocytes modulates the activity of osteoblasts, further promoting bone formation.

## 2. Results

### 2.1. CB Promotes Bone Development in Layer Pullets

We supplemented hens with CB by oral gavage and examined bone development. CB significantly improved the tibial index (*p* < 0.05) and increased the ash (*p* < 0.05) and calcium content (*p* < 0.05) of the tibia, but no significant change was seen in the bending strength of the tibia (*p* > 0.05) (Figure 1A). We explored why the flexural strength did not change and found that feeding with CB increased bone mineral density (BMD) on the whole and in both ends of the tibia (*p* < 0.05) but not in the mid-segment (i.e., the bending strength testing site) (*p* > 0.05) (Figure 1B). The results of H&E staining of the tibia revealed that CB increased the number and thickness of tibial trabecular bones (Figure 1C), indicating that CB regulates tibial bone development in layer pullets.

### 2.2. CB Promotes Osteoblast Metabolism

The bone mass of layer pullets is regulated by the bone resorption of osteoclasts and the bone formation of osteoblasts [26]. The numbers of tartrate-resistant acid phosphatase (TRAP)-stained osteoclasts were lower in the tibial sections of CB groups (Figure 2A). However, CB did not alter the mRNA expression of receptor activator of nuclear factor kappa-Β ligand (RANKL) in tibia (*p* > 0.05), nor did it alter osteoprotegerin (OPG) (*p* > 0.05) (Figure 2B), both of which are critical regulators of osteoclastogenesis and bone resorption [27]. Alkaline phosphatase (ALP) is a metabolic marker of osteoblasts [28]; administration of CB upregulates ALP-stained areas in tibial sections (Figure 2A) and increases serum levels of ALP in layer pullets (*p* < 0.05) (Figure 2C). Measurement of transcript enrichment in osteoblast differentiation—including runt-related transcription factor 2 (Runx2) (*p* < 0.05), which is essential for osteoblast differentiation and chondrocyte maturation [29], and bone morphogenetic protein 2 (BMP2) (*p* > 0.05), which has important roles in bone remodeling and homeostasis [30]—revealed that CB treatment amplified osteoblast differentiation in tibias (Figure 2D). CB treatment had no significant effect on the relative expression of RANKL, OPG, or Runx2 in the gut and femur (*p* > 0.05), but it did promote the expression of BMP2 (*p* < 0.05) (Appendix A).

### 2.3. CB Upregulates the Proportion of Treg Cells

Gut microbiota and bone development are closely related. To assess whether probiotic administration significantly modulated intestinal microbiota diversity in layer pullets and whether a difference in microbiota diversity could account for the role of CB in bone formation, we extracted DNA from colonic microbiota and performed 16S rRNA sequencing. The results showed that CB treatment had no significant effects on the diversity of gut flora, microbial community structure, microbial species composition, or SCFA concentrations in the gut and serum of pullets (Appendix A). These data suggest that treatment with CB does not cause broad changes in the bacterial communities in layer pullets.

Acting on the immune system is another potential pathway through which CB may affect bone metabolism. The immune and skeletal systems share a variety of molecules, including cytokines, chemokines, hormones, receptors, and transcription factors [31]. The expressions of the bone metabolism-related cytokines *IL-1*, *IL-6*, *IL-10*, *tumor necrosis factor alpha* (*TNF-α)*, and *nuclear factor kappa-light-chain-enhancer of activated B* (*NF-κB)* cells produced by each immune organ were detected after CB treatment. CB upregulated the expression of *IL-10* mRNA in the tibia, cecum, and spleen (*p* < 0.05); the expression of *IL-1* in the spleen (*p* < 0.05); and the expression of *NF-κB* in the cecum (*p* < 0.05) (Figure 3A). *IL-10* is considered an important regulator of bone homeostasis [32]; it activates and is a key mediator of Treg cells.

Foxp3 is a hallmark molecule of Treg cells, but a chicken ortholog was only identified in 2022, and its function is unclear [33,34]. Avian Treg cells are usually defined as CD4^+^CD25^+^ cells [35]; here, we used CD3^+^CD4^+^CD25^+^ as the standard for isolation and identification of avian Treg cells (Figure 3B). A gavage test found that CB treatment could significantly upregulate the proportion of Treg cells in the spleen (*p* < 0.05) and peripheral blood (*p* < 0.0001) lymphocytes but had no significant effect on the proportion of Treg cells in bone marrow (*p* > 0.05) (Figure 3C). Additional experiments were performed using peripheral blood lymphocytes cultured in vitro, and we found that the culture supernatant of CB could increase the proportion of Treg cells in vitro (*p* < 0.01) (Figure 3C). The expression of *IL-1* in 0.3% CB-treated (*p* < 0.01) or 0.6% CB-treated (*p* < 0.001) lymphocytes was significantly increased, while the mRNA expressions of *IL-6*, *IL-10*, *TNF-α,* and *RANKL* did not change significantly (*p* > 0.05) (Figure 3D).

### 2.4. CB Promotes Osteoblast Activity In Vitro but Requires Lymphocyte Involvement

To conclusively demonstrate the effect of CB on osteoblasts, we evaluated the effects of CB and SB (after the SCFA content in the culture supernatant of CB was analyzed with GC-MS, the same amount of SCFAs was used as a control treatment) on the bone-forming ability of cultured osteoblasts (Figure 4A). In these experiments, CB culture supernatant and SB were added to the osteoblast induction system; all osteoblasts were treated with control (Con), CB, or SB for 8 days. CB and SB had no significant effects on the mineral nodule formation ability and ALP generation ability of osteoblasts (*p* > 0.05) (Figure 4B). Subsequently, the lymphocyte culture supernatant (LCon) alone and the culture supernatants of lymphocytes previously treated with CB (LCB) or with SB (LSB) were added (Figure 4A). It can be seen that the LCon treatment enhanced the formation of mineral nodules in osteoblasts and ALP production (*p* < 0.05), while LCB and LSB induced higher levels of osteogenic activity (*p* < 0.01) (Figure 4B). However, the bone formation indexes of the LCB and LSB groups showed no significant changes compared to the LCon group (*p* > 0.05) (Figure 4B). 

With regard to the detection of key gene expressions of bone metabolism-related signaling pathways in osteoblasts after treatment, it should first be noted that the RANKL/OPG pathway is a key pathway through which osteoblasts regulate osteoclast activity. Our experiments showed that the LCon, LCB, and LSB treatments all decreased the mRNA relative expression of *RANKL* compared to the Con group (*p* < 0.05), but no differences among the three groups were found (*p* > 0.05) (Figure 4C). Compared to the Con group, the LCon, LCB, and LSB treatments all increased *OPG* (*p* < 0.05); LCB and LSB were more inducible than LCon (*p* < 0.05), and LSB was more inducible than LCB (*p* < 0.05) (Figure 4C). 

The RUNX2/BMP2 pathway is another key pathway for osteoblast maturation and differentiation. The experiments showed that LCon, LCB, and LSB treatments all increased *BMP2* compared to the Con group (*p* < 0.05), but no differences among the three groups were found (*p* > 0.05) (Figure 4D). Only the LCB treatment significantly increased the expression of *RUNX2* (*p* < 0.01) (Figure 4D). 

The WNT signaling pathway is one of the central signaling pathways in the regulation of bone development, homeostasis, and bone mineral density. *Wnt3a* can effectively induce ALP activity in osteoblasts and promote osteoblast differentiation [36], as well as upregulate the expression of *OPG* in osteoblasts to inhibit osteoclast formation [37]. *Wnt5a*, a non-canonical WNT ligand, is strongly expressed in osteoblasts, and the *Wnt5a* secreted by osteoblasts enhances the expression of receptor activator of nuclear factor κ B (Rank) in osteoclast precursors and induces osteoclast differentiation [38]. Our experiments showed that the LCon, LCB, and LSB treatments all increased *Wnt3a* compared to the Con group (*p* < 0.05), and LCB and LSB were more inducible than LCon (*p* < 0.05) (Figure 4E). The LCon, LCB, and LSB treatments all decreased *Wnt5a* compared to the Con group (*p* < 0.05), but no differences among the three groups were found (*p* > 0.05) (Figure 4E).

## 3. Discussion

We report that oral administration of CB to pullets in the brooding stage increased tibia development due to the stimulation of bone formation. We also show that CB supplementation increased the number of Treg cells in peripheral blood and the spleen (Treg cells are a type of bone metabolism regulatory cell that are essential for bone development). These findings demonstrate that CB may represent a therapeutic strategy to enhance bone anabolism. 

Growing evidence suggests that the gut microbiome is a key regulator of bone health, and the concept of a gut–bone axis has been proposed [39]. The present study showed that CB can significantly promote the development of tibia in layer pullets. Interestingly, the significantly increased bone index and bone mineral content did not improve the flexural strength of the bone, and further BMD results revealed that the cause of the problem may have been that the effects of CB were concentrated in the cancellous region of the tibia. Changes in the number and density of tibial trabecular bones in the CB-treated group also verified this conjecture. 

Bone is regulated by both osteoblasts and osteoclasts [26]. RANKL is a key factor in osteoclastogenesis; RANKL and RANK combine to mediate osteoclastogenesis and signaling, while OPG inhibits osteoclastogenesis by preventing RANKL–RANK interaction [40]. Mature osteoblasts deposit a mineralized matrix and begin to express ALP, thus serving as a major marker of the bone formation process [28]. RUNX2 is expressed in uncommitted mesenchymal cells; it directs the differentiation of mesenchymal stem cells into osteoblasts and plays an important role in indicating bone formation [41]. This study found that CB increased serum ALP levels and tissue *RUNX2* mRNA levels but did not affect the expression of *RANKL* and *OPG*. Therefore, we concluded that CB exercises its bone development-promoting activity by regulating the activity of osteoblasts.

It is well-known that the gastrointestinal tract of poultry contains an extremely diverse and complex microbiota, and feeding chickens with probiotics can alter the composition of the intestinal microbiota through effects on the production of organic acids and competition for nutrients in the microbiota [42]. CB can influence the host through several means, including modifying the gut microbiome composition and function and secreting factors that directly influence host cells. Even though our recent research suggested that CB is able to alter microbial communities in a model of *Salmonella* infection [25], taxonomy data suggest that, at the studied concentration (enough to promote bone development), CB does not modify broad, phylum-level bacterial community composition. 

A close connection exists between the skeletal system and the immune system [43,44]. Recent studies have described the immune effects of probiotics, including the inhibition of proinflammatory cytokine expression [45,46] and the stimulation of Tregs [47]. It has been reported that dietary *C. butyricum* activates the expression of *IL-10* and reduces the expression of the proinflammatory cytokines *IL-1β* and *TNF* to modulate the inflammatory response [48,49]. Li’s study found that CB significantly decreased the high stocking density-induced expression levels of *IL-1β* and *TNF-α* in the ileum of broilers at different stages [17]. A previous study conducted by our group found that CB reduced the elevation of *IL-1* and *IL-6* caused by *Salmonella* infection in broilers [23,24]. In contrast, this study used animals in healthy conditions and found no changes in *IL-1* or *IL-6* expression. *IL-10* is a potent anti-inflammatory cytokine that suppresses immune proliferation and inflammatory responses and is, therefore, also considered to be an important regulator of bone homeostasis [50]. *IL-10* also directly inhibits osteoclast formation [51] and enhances osteoblast differentiation [52]. CB increases *IL-10* expression in mice [48] and in broilers [53] under enteritis conditions. Our results showed that CB treatment increased *IL-10* expression in various organs, including the tibia, spleen, and cecum, despite healthy conditions. In vitro, CB significantly promoted the expression of *IL-1* mRNA in lymphocytes, which may have been related to the activation and proliferation of lymphocytes. However, in vitro experiments did not show an increase in IL-10 mRNA expression in lymphocytes after CB treatment, which means that a more complex mechanism is required for CB treatment to induce *IL-10* in vivo.

Tregs are a major source of *IL-10* in the gut [54]. Early reports described the skeletal effects of Treg cells, including inhibition of osteoclast production [31] and enhancement of osteoblast activity [55]. Existing research has shown that butyrate [56] and CB [57] can induce Treg cell production, and our previous study confirmed that CB can regulate the proportions of Tregs in mice [58]. The present study found that CB treatment increased the proportions of Tregs in the spleen, peripheral blood, and cultured lymphocytes but did not affect their proportions in bone marrow. It can be suggested that CB can positively regulate Tregs, which are closely related to bone metabolism in layer pullets, but whether it affects bone metabolism through this regulation needs to be further explored.

The link between lymphocytes and the bone anabolic activity of CB and SB was established using an in vitro model. Osteoblasts were treated with a culture supernatant of CB or a culture supernatant of lymphocytes pre-treated with CB. Chen’s study found that treatment of osteoblasts with a culture supernatant of butyric acid-producing bacteria (*Lactobacillus acidophilus*) and sodium butyrate stimulated the proliferation, differentiation, and maturation of osteoblasts [59]. Similarly, Schroeder found that histone deacetylase inhibitors could promote osteoblast maturation [60]; butyric acid produced by CB is a histone deacetylase inhibitor. Our experimental results contrast with these earlier results: we found that the addition of CB or butyrate alone did not have any effect on the osteogenic ability of osteoblasts in vitro, but the use of lymphocyte culture supernatant alone could promote the osteogenic ability of osteoblasts in vitro, with LCB and LSB being able to induce higher levels of bone formation on this basis. These findings suggest that the beneficial effects of CB on osteoblastic bone formation depend on lymphocytes. 

This inference was validated by the present study’s investigation of key pathways related to bone metabolism (RANKL/OPG, RUNX2/BMP2, and WNT) in osteoblasts. The results showed that, in addition to the expression of *RUNX2*, all treatment groups containing a lymphocyte culture supernatant could positively regulate the expression of key genes in these pathways. The relative expressions of *RANKL*, *BMP2*, and *Wnt5a* were not significantly altered by the addition of CB or SB relative to the lymphocyte supernatant group where lymphocytes played a dominant role. For *OPG* and *Wnt3a*, further addition of CB or SB showed stronger induction. The results show that lymphocytes played a key role in the bone formation ability of the cultured pullet osteoblasts in vitro, and CB could further stimulate osteoblast bone formation by regulating the expression of *OPG*, Wnt3a, and *RUNX2* with the mediation of lymphocytes. At the same time, we observed that the effects of LCB and LSB treatments on *OPG* and *RUNX2* in osteoblasts were not completely consistent: LSB had a stronger effect on promoting *OPG*, while LCB had a stronger effect on promoting *RUNX2*. The acetic acid and butyric acid contents in the LSB and LCB groups were exactly the same, indicating that the effect of CB may not only depend on butyric acid but may also have other means of action.

## 4. Materials and Methods

All procedures used in this study were approved by the Animal Care Committee of Shandong Agricultural University (P. R. China) and carried out in accordance with the guidelines for experimental animals published by the Ministry of Science and Technology (Beijing, P. R. China).

### 4.1. Bacterial Strains

CB was obtained from Dalian Sanyi Animal Medicine Company (Dalian, China). The strain was cultured anaerobically with Reinforced Clostridial Medium (Qingdao Haibo Biological, Qingdao, China) broth at 37 °C for 48 h. The bacterial concentration was adjusted to 2 × 10^8^ CFU/mL.

Preparation of CB culture supernatant: For the preparation, 1 × 10^8^ CFU of CB was inserted into a 5 mL Reinforced Clostridial Medium (RCM) liquid anaerobic culture at 37 °C and 220 rpm for 48 h; it was centrifuged at 500× *g* for 5 min, then the supernatant was filtered with a 0.22 μm filter.

Butyrate control: The results concerning the content of each short-chain fatty acid in the culture supernatant of *Clostridium butyricum* are shown in Table 1. Sodium acetate and sodium butyrate were used to prepare a mixture of sodium acetate and sodium butyrate.

### 4.2. Experimental Design

The experiment adopted a single-factor, completely randomized design. The researchers selected 80 one-day-old Hailan brown layer pullets with similar body weights and randomly divided them into two treatment groups; each group had four replicates, and each replicate had ten chickens. After the start of the experiment, the Con group was given oral saline at 0.5 mL/2 days, while the CB group was given an oral solution of 2 × 10^8^ CFU/mL of CB at 0.5 mL/2 days. The CB used in the experiment employed physiological saline as a solvent. The rearing environment corresponded with actual production requirements, and immunization was carried out according to the routine immunization program. Free access to food (Table 2) and water was provided. At 21 days old—i.e., the end of the experiment—all chickens were fasted overnight, and eight chickens from each treatment were randomly selected. After blood samples were taken from a wing vein, the chickens were euthanized via cervical dislocation. After the chickens were dissected, the spleen, cecal contents, femur, and tibia were weighed and sampled. The tissue samples were immediately snap-frozen in liquid nitrogen and stored at −80 °C for further analysis. Serum was separated using centrifugation at 1500× *g* for 15 min and stored at −20 °C until analysis. In addition, eight chickens were randomly selected from each treatment and euthanized via cervical dislocation, after which their peripheral blood, spleen, and bone marrow were collected for flow cytometry.

### 4.3. Bone Function Evaluation Index

Organ index: The organ index was calculated as the organ weight divided by the chicken weight.

Ash and calcium content: The tibia samples were treated with a mixture of alcohol and benzene at a ratio of 2:1 for 96 h for degreasing and then dried at 105 °C to maintain weight. The degreased bone samples were used for the measurement of calcium content. Bone calcium content was measured according to a method described previously [61]. Briefly, the degreased bones were first burned in a crucible heated by an electric ceramic furnace until they were carbonized, and then they were burned in a muffle furnace at 550 °C for 6 h; the product was bone ash. The calcium content was then determined using the potassium permanganate method.

Bone bending strength: Bone bending strength was determined with a method described previously [62]. In simple terms, the three-point bending test was used to detect the maximum stress a bone could withstand before it broke.

BMD: A bone densitometer (InAlyzer, Baitai Technology Co., Ltd., Guangzhou, China) was used to measure bone mineral density at different spots in the tibia and femur. High-energy and low-energy two-layer X-rays were emitted through the X-ray tube; the attenuation of the two layers of rays in different tissues differed, enabling the corresponding BMD to be obtained through software calculations (Appendix A).

Bone histological analysis: The tibia samples were decalcified in 12% ethylenediaminetetraacetic acid (EDTA) for 3 weeks and then embedded in paraffin. Histological sections were prepared for ALP, TRAP, and H&E staining. The specimens were then examined and photographed under a high-quality microscope.

Serum ALP: Serum concentrations of ALP were measured using commercial kits (Jiancheng Bioengineering Institute, Nanjing, China). All procedures were conducted according to the manufacturers’ instructions.

### 4.4. Real-Time PCR Analyses

Total RNA was extracted with RNAiso Plus Reagent (TransGen Biotech, China) and reverse-transcribed into complementary DNA (cDNA) using the RT First Strand cDNA Synthesis Kit (Roche, Germany). Quantitative real-time PCR (qRT-PCR) was performed using an ABI QuantStudio 5 PCR machine (Applied Biosystems; Thermo, Waltham, MA, USA) at 95 °C for 10 min, followed by 40 cycles of 95 °C for 15 s and 60 °C for 30 s, with 2×SYBR Green qPCR Master Mix (Servicebio, Wuhan, China). The primer sequences are provided in Table 3. Relative mRNA levels of specific genes were quantified using the 2−ΔΔCt method values with respect to the values for *β-actin*. 

### 4.5. Bone RNA Extraction

The bone tissue sample (upper quarter of the tibia) was cut, ground in liquid nitrogen, placed in a 1.5 mL centrifuge tube to which 1 mL TransZol was added, and homogenized with a tissue homogenizer in an ice bath. Then, it was centrifuged at 12,000 rpm for 15 min at 4 °C, and the supernatant was transferred to a new 1.5 mL centrifuge tube. Next, 200 μL of ice chloroform was added, the mixture was vigorously shaken for 15 s, placed on ice for 20 min, and then centrifuged at 12,000 rpm and 4 °C for 20 min. The supernatant was transferred to a new 1.5 mL centrifuge tube, 250 μL of pre-cooled isopropanol and 250 μL of high salt solution (0.8 mol/L sodium citrate and 1.2 mol/L sodium chloride) were added, and the sample was placed on ice for 20 min. It was then centrifuged at 12,000 rpm and 4 °C for 15 min, and the supernatant was discarded. One milliliter of pre-cooled 75% ethanol was added to wash the RNA pellet, and then centrifugation was applied at 7500 rpm for 2 min at 4 °C. This was repeated three times, and then the centrifuge tube was allowed to stand until the ethanol evaporated. The RNA was then dissolved in DEPC water and the sample stored at −80 °C until use.

### 4.6. Gut Microbiome Analysis and Determination of SCFA Concentrations

Serum and cecal contents collected from chickens in the Con or CB groups were used for gut microbiome analysis and determination of the SCFA concentration as described previously [58]. 16S rRNA sequencing was performed at Shanghai Personal Biotechnology Co., Ltd. (Shanghai, China). DNA was extracted from the cecal content by using a Stool DNA kit (Omega Bio-Tech, Doraville, CA, USA). The V4–V5 region of the bacterial 16S rRNA gene was amplified with PCR. The PCR products were collected and sequenced using the Illumina MiSeq2000 platform. Sequences were analyzed with the QIIME software package and UPARSE pipeline. The high-quality sequences were clustered into operational taxonomic units (OTUs) with 97% identity.

The SCFA (acetate, propionate, butyrate, pentanoic acid, and isovalerate) concentrations were determined by using a gas chromatography–mass spectrometer system (ThermoFisher, USA). The cecal content was diluted, acidified, and extracted ultrasonically on ice for 10 min. The samples were then centrifuged at 4000 rpm and 4 °C for 20 min. The cecal supernatant or serum was injected into a GCMS ISQ LT (ThermoFisher) and TRACE GCMS ISQ LT (ThermoFinnigan, San Jose, CA, USA) with the following parameter settings: column temperature: 100 °C (5 min)–5 °C min^−1^–150 °C (0 min)–30 °C min^−1^–240 °C (30 min); flow rate: 1 mL min^−1^; split ratio: 75:1; carrier gas: helium; column: TG WAX 30 m × 0.25 mm × 0.25 µm; injector: 240 °C; mass spectrometry: EI source; bombardment voltage: 70 eV; single-ion scanning mode: quantitative ions 60 and 73; ion source temperature: 200 °C; cable temperature: 250 °C. The external standard curve method was used for quantitative analysis.

### 4.7. In Vitro Culture and Processing of Lymphocytes

Lymphocyte isolation: The instructions for the lymphocyte isolation kit (Tianjin Haoyang Biological Products Technology Co., Ltd., Tianjian, China) were followed to isolate lymphocytes from the spleen, bone marrow, and peripheral blood. The separation method for the peripheral blood can be used as an example: the peripheral blood of the test chicken was collected, carefully added to the liquid level of the lymphocyte separation solution, and centrifuged at 1500× *g* and 4 °C for 20 min. After the centrifugation, the tube was divided into four layers, with the second layer (milky white mist layer) comprising the lymphocytes; the lymphocytes were washed twice with a washing solution.

Lymphocyte culture: The isolated lymphocytes were adjusted to 1 × 10^6^ cells/mL, and 1.8 mL of RPMI 1640 medium containing 10% FBS, 5 mM LPS (L7770-1MG, Sigma-Aldrich, St. Louis, MO, USA), and 5 mM PHA-L (11249738001, Sigma-Aldrich, St. Louis, MO, USA) was added to each well of a six-well plate; the lymphocytes were cultured at 37 °C in a 5% CO_2_ incubator.

Lymphocyte treatment: Lymphocytes were cultured for 12 h and then treated with 200 μL of the diluted CB culture supernatant or SB for 48 hours (the overall CB culture supernatant or SB concentration was 0.3%). Lymphocytes were collected for the detection of relevant gene expressions and lymphocyte subset ratios, and lymphocyte culture supernatants were collected for osteoblast treatment.

### 4.8. In Vitro Culture and Processing of Osteoblasts

Bone marrow mesenchymal stem cell isolation: Zhou’s method [63] was employed for the isolation with appropriate modifications. The tibias of 18 embryo-age chicks were isolated and washed three times with Hanks’ Balanced Salt Solution (pre-warmed at 37 °C, double antibody) to remove the muscle and mucosal tissue on the bone. A syringe was drawn with DMEM medium (10% FBS, double antibody) to rinse the tibia bone marrow cavity, a process repeated in triplicate. The cell suspension was filtered with a 75 μm cell strainer, and 2 mL was added to each well of a six-well plate and placed in a cell incubator at 37 °C with 5% CO_2_; the medium was replaced every 2 days.

Osteoblast induction: Li’s method [64] was employed for the osteoblast isolation and culture with appropriate adjustments. Bone marrow mesenchymal stem cells were cultured to 80% confluence; the original medium was discarded and replaced with osteogenic induction medium (50 μg/mL L-ascorbic acid, 2 mM β-gly, and 0.01 μM dexamethasone were added to the original medium). The medium was changed every 2 days, with each change involving rinsing with 37 °C pre-warmed PBS, over a total of 10 days of induction.

Osteoblast treatment: On the fifth day of induction, the culture supernatant of CB (0.3%), SB, lymphocyte culture supernatant, and pre-treated (CB and SB) lymphocyte culture supernatant were added to the original induction solution.

### 4.9. Flow Cytometry

The following anti-chicken antibodies were used for cell surface staining: Mouse Anti-Chicken CD3-Pacific Blue (SBA-8200-26), Mouse Anti-Chicken CD4-PE (SBA-8210-09), Mouse Anti-Chicken CD8α-APC (SBA-8220-11) (Southern Biotech, Birmingham, AL, USA), and Human Anti-Chicken CD25-FITC (HCA173F) (Bio-Rad, Hercules, CA, USA). Flow cytometry was performed on a LSR Fortessa4 system (BD Biosciences, Franklin Lakes, NJ, USA), and the data were analyzed using the FlowJo software (Tree Star, Inc., Ashland, OR, USA) (Appendix A).

### 4.10. Statistical Analysis

The data are expressed as means ± SD. The results were analyzed using one-way ANOVA in the Statistical Analysis Systems statistical software package (Version 8e; SAS Institute Inc., Cary, NC, USA). Differences between means were evaluated using Duncan’s significant difference tests. Means were considered significant at *p* < 0.05.

## 5. Conclusions

In conclusion, this study showed that CB administration could significantly upregulate proportions of Treg cells in the spleen and peripheral blood of layer pullets and the relative expression of *IL-10*; furthermore, it could also significantly increase the tibial index, tibial mineral content, and tibial bone of layer pullets. In vitro experiments proved that CB promotes the osteogenic activity of osteoblasts in vitro (via RANKL/OPG, RUNX2/BMP2, and WNT signaling pathways); however, it requires the participation of lymphocytes. These findings contribute to a deeper understanding of the underlying mechanisms through which CB improves bone development and immune homeostasis in layer pullets. However, this study did not confirm that CB regulates the bone metabolism of pullets through Treg cells, and further mechanistic validation through Treg cell depletion experiments, as well as large scale clinical studies, is needed to validate their connections to bone.

## Figures and Tables

**Figure 1 ijms-24-01457-f001:**
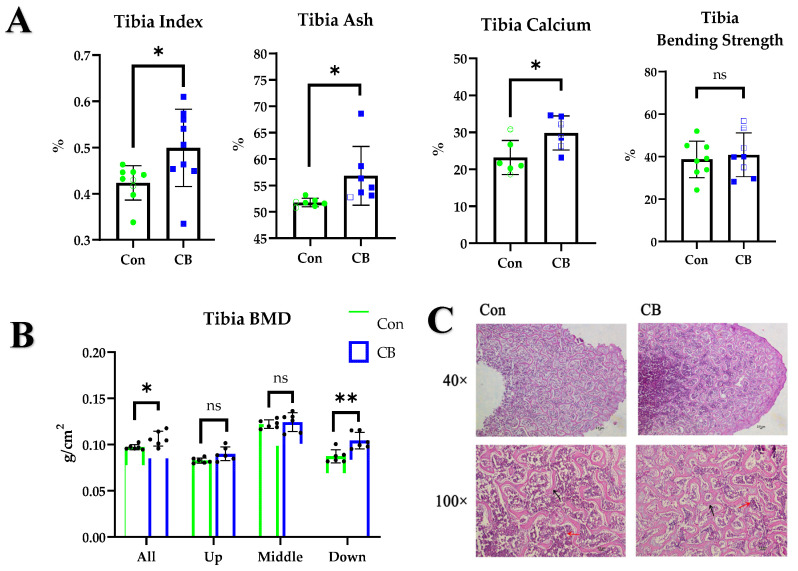
Effects of *Clostridium butyricum* (CB) supplementation on tibia development in layer pullets. (**A**) Tibial index, tibial ash, tibial calcium content, and tibial flexural strength. (**B**) Bone mineral density (BMD) in different areas of the tibia. (**C**) Histomorphology of the tibia. Con: normal saline; CB: Clostridium butyricum. The data are presented as the mean ± SD. *, *p* < 0.05, **, *p* < 0.01, ns, no difference.

**Figure 2 ijms-24-01457-f002:**
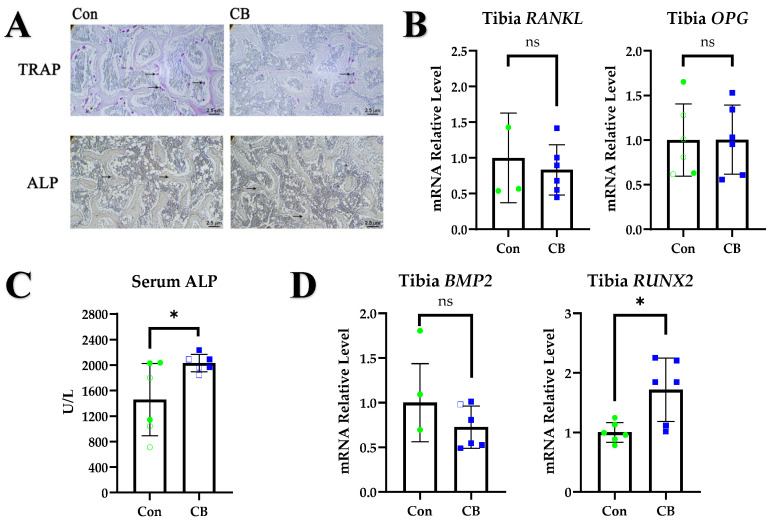
Effects of CB on osteoblasts and osteoclasts. (**A**) Tibial sections were stained for tartrate-resistant acid phosphatase (TRAP) and alkaline phosphatase (ALP). Scale bar = 2.5 µm. (**B**) Tibial mRNA expression of *receptor activator of nuclear factor kappa-Β ligand* (*RANKL)* and *osteoprotegerin (OPG)*. (**C**) ALP levels in serum. (**D**) Tibial mRNA expression of *bone morphogenetic protein 2* (*BMP2)* and *runt*-*related transcription factor 2* (*RUNX2)*. Con: normal saline; CB: *Clostridium butyricum*. The data are presented as the mean ± SD. *, *p* < 0.05, ns, no difference.

**Figure 3 ijms-24-01457-f003:**
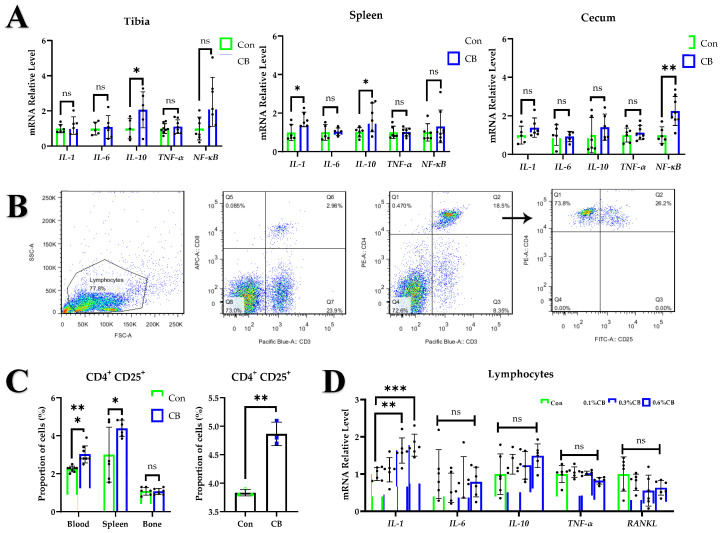
The effects of CB on the immune system. (**A**) Relative mRNA expression levels of *IL-1*, *IL-6*, *IL-10*, *tumor necrosis factor alpha* (*TNF-α)*, and *nuclear factor kappa-light-chain-enhancer of activated B cells (NF-κB)* in the tibia, spleen, and cecum. (**B**) Gating strategy for identification of T regulatory (Treg) cells with flow cytometry. (**C**) Proportion of Treg cells in the peripheral blood, spleen, bone marrow, and lymphocytes cultured in vitro. (**D**) Relative mRNA expression levels of IL-1, *IL-6*, *IL-10*, *TNF-α*, and *RANKL* in lymphocytes cultured in vitro. Con: normal saline; CB: *Clostridium butyricum*; SB: mixture of sodium acetate and sodium butyrate. The data are presented as the mean ± SD. *, *p* < 0.05, **, *p* < 0.01; ***, *p* < 0.001, ns, no difference.

**Figure 4 ijms-24-01457-f004:**
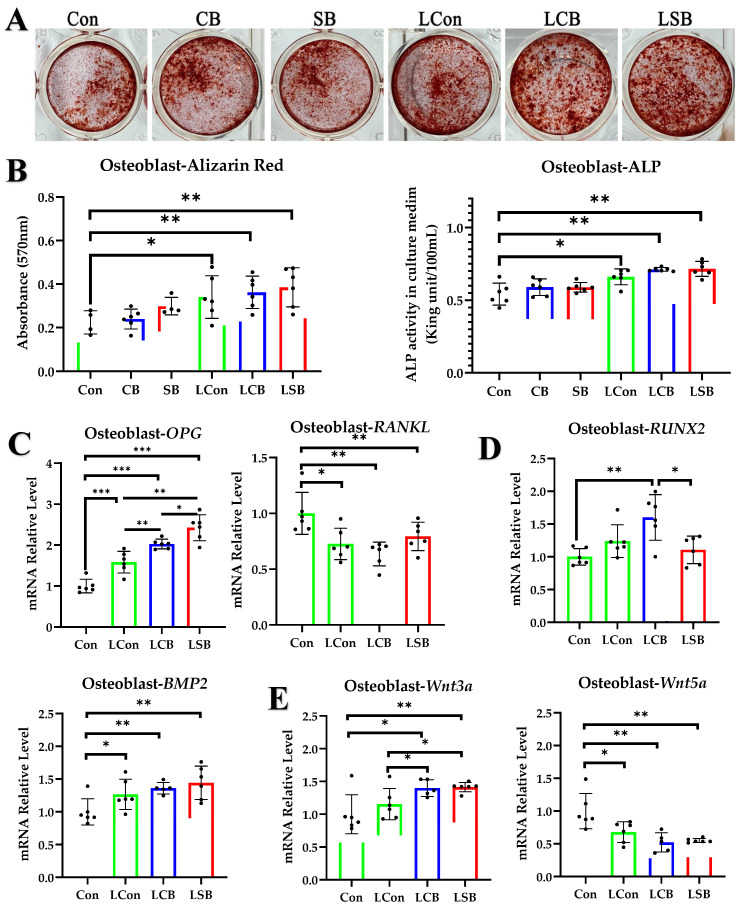
The effect of CB on osteoblasts. (**A**) Alizarin Red staining of osteoblast mineral nodules. (**B**) Alizarin Red staining quantification and ALP levels in culture supernatants. (**C**–**E**) Relative mRNA expression levels of *RANKL*, *OPG*, *RUNX2*, *BMP2*, *Wnt3a*, and *Wnt5a* in osteoblasts cultured in vitro. Con: normal saline; CB: *Clostridium butyricum*; SB: mixture of sodium acetate and sodium butyrate; LCon: lymphocyte control; LCB: lymphocytes pretreated with CB; LSB: lymphocytes pretreated with SB. The data are presented as the mean ± SD. *, *p* < 0.05, **, *p* < 0.01; ***, *p* < 0.001.

**Table 1 ijms-24-01457-t001:** Results for CB culture supernatant short-chain fatty acid contents.

Test Items	Test Result	Unit	Test Basis
Acetic acid	247.27	μg/g	GCMS
Propionic acid	Not detected	μg/g	GCMS
Isobutyric acid	Not detected	μg/g	GCMS
n-Butyric acid	866.50	μg/g	GCMS
Isovaleric acid	Not detected	μg/g	GCMS
n-Valeric acid	Not detected	μg/g	GCMS

**Table 2 ijms-24-01457-t002:** Composition of experimental diet.

Items	Composition, %
Ingredients	
Corn	63
Soybean meal	24
Limestone	8
NaCl	0.3
Choline chloride (50%)	0.1
Premix ^1^	4.6%
Total	100
Nutrient levels ^2^	
Metabolizable energy (Kcal/kg)	2700
Crude protein (%)	16.1
Lysine (%)	0.79
Methionine (%)	0.34
Calcium (%)	3.50
Available phosphorus (%)	0.45

^1^ The vitamin and mineral premix used the following quantities per kilogram of diet: vitamin A, 8800 IU; vitamin D3, 3300 IU; vitamin K, 2.2 mg; vitamin E, 16.5 IU; cholecalciferol, 2800 IU; riboflavin, 18 mg; niacin, 50 mg; pantothenic acid, 28 mg; biotin, 0.1 mg; folic acid, 0.6 mg; iron, 55 mg; selenium, 0.3 mg; copper, 5.5 mg; zinc, 88 mg; iodine, 1.7 mg; manganese, 88 mg; calcium, 5.7 g; and phosphorus, 3.3 g. ^2^ Nutrient levels are all calculated values.

**Table 3 ijms-24-01457-t003:** Primers for the qRT-PCR used in this study.

Gene Name	Genbank Number	Primer Position	Primer Sequences (5′→3′)
*IL-1*	Y15006	Forward	ATCACAGCCACACAGAAGACG
		Reverse	TGACTTTCCCCACAGCCTTA
*IL-6*	NM204628.2	Forward	CTCCTCGCCAATCTGAAGTC
		Reverse	AGGCACTGAAACTCCTGGTC
*IL-10*	NM204100.1	Forward	CGCTGTCACCGCTTCTTCA
		Reverse	TCCCGTTCTCATCCATCTTCTC
*TNF-α*	AY765397	Forward	CATTTGGAAGCAGCGTTTGG
		Reverse	GGTTGTGGGACAGGGTAGGG
*NF-κB*	NM205129	Forward	CTCTCCCAGCCCATCTATGA
		Reverse	CCTCAGCCCAGAAACGAAC
*RANKL*	NM001083361.2	Forward	TGTTGGCTCTGATGCTTGTC
		Reverse	TCCTGCTTCTGGCTCTCAAT
*OPG*	DQ098013.1	Forward	CGCTTGTGCTCTTGGACATT
		Reverse	GCTGCTTTACGTAGCTCCCA
*BMP2*	NM001398170.1	Forward	CCTTCGGAAGACGTCCTCAG
		Reverse	CTGAGTGCCTGCGGTACAGA
*RUNX2*	NM204128.1	Forward	TTTTTCCTGCCCGTATTCTG
		Reverse	GCTTGGTGCTGGAGAGTCTT
*Wnt3a*	EF068232.1	Forward	GTGGCTTTTGCAGTGACCAG
		Reverse	GTTGTGCCTTCATGGCTG
*Wnt5a*	NM001037269.1	Forward	TGGCTTCTCAGTACCTCGTAGTGG
		Reverse	GCCGAAGACGGACGTGTTGTC
*β-actin*	L08165	Forward	GAGAAATTGTGCGTGACATCAAGG
		Reverse	CACCTGAACCTCTCATTGCCA

## Data Availability

The datasets produced and/or analyzed during the current study are available from the corresponding author upon reasonable request.

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
