# Peer review of "Clostridium butyricum Can Promote Bone Development by Regulating Lymphocyte Function in Layer Pullets"

_ijms, 2023, doi:10.3390/ijms24021457_

Round 1
Reviewer 1 Report (Previous Reviewer 2)
The manuscript has been sufficiently improved with respect to the previous comments of all the reviewers.
Author Response
Dear Reviewer,
Thank you so much for your letter and comments on our manuscript. We appreciate very much for your effort and the valuable suggestions and comments.
With best regards,
Prof. Dr. Shuhong Sun
Reviewer 2 Report (Previous Reviewer 3)
The corrections have been made properly and the methodology is improved. I have no more comments.
Author Response
Dear Reviewer,
Thank you so much for your letter and comments on our manuscript. We appreciate very much for your effort and the valuable suggestions and comments.
With best regards,
Prof. Dr. Shuhong Sun
Reviewer 3 Report (New Reviewer)
The manuscript entitled "Clostridium butyricum can promote bone development by regulating lymphocyte function in layer pullets" examined the impact of CB supplementation on bone and immune homeostasis. The overall study is interesting, and the authors conducted various experiments to prove their claims. However, the only suggestion is authors should mention the impact of CB on signaling pathways related to bone development in the conclusion section. Additionally, the conclusion section should also consist of the drawbacks of the current studies and future perspectives.
Author Response
Dear Editors,
Thank you so much for your letter and the comments concerning our manuscript. We appreciate very much for your effort and the valuable suggestions and comments. All comments and suggestions were fully considered and the manuscript was revised. We hope the manuscript is suitable for publication.
I am looking forward to hearing from your positive decision.
With best regards,
Prof. Dr. Shuhong Sun
Reviewer:
Minor points:
Q1. However, the only suggestion is authors should mention the impact of CB on signaling pathways related to bone development in the conclusion section.
Response: We greatly thank the comment. The necessary information has been supplemented (Line 478-479).
Q2. Additionally, the conclusion section should also consist of the drawbacks of the current studies and future perspectives.
Response: We greatly thank the comment. The necessary information has been supplemented (Line 482-484).
This manuscript is a resubmission of an earlier submission. The following is a list of the peer review reports and author responses from that submission.
Round 1
Reviewer 1 Report
In this paper, the authors found that CB administration significantly increased the percentage of Treg cells and the relative expression of IL-10 in the spleen and peripheral blood of chickens, significantly increased bone mineral content and bone mass, and promoted osteogenic activity of osteoblasts in vitro. They argued that CB improves skeletal development and immune homeostasis. The conclusions of this paper are consistent with the well-established finding that CB administration increases IL10 production, which in turn promotes offspring growth. However, the in vivo results are largely predictable against a background of established mechanisms and seem to lack novelty. In the mechanism of increased bone tissue, increased BMP2 and Wnt3a production were found as a mechanism for increased osteoblast differentiation, which is also expected, but the direct effect of IL10 on osteoblasts was not examined, so the relationship with IL10 that the authors claim is largely speculative and unproven.
Major
1. First, CB induces osteoblast differentiation but decreases RANKL expression. In general, when osteoblast differentiation is promoted, RANKL expression in osteoblasts should increase. Therefore, the lack of experimental results to support the claim that CB induces osteoblast differentiation but decreases RANKL expression is very strange. Why?
It seems llikely that the mechanisms may be more complexed.
2. To prove the promotion of osteoblast differentiation, it is necessary to accurately assess the increase in expression of osteoblast markers over time. In the present study, only one point was measured, which does not prove that differentiation was promoted.
3. There is already established evidence of the effects of CB on the gut microbiota and its effects on animal growth。However, it has not been verified whether these effects are replicated in this study, so it is not possible to verify whether CB administration in this study was appropriate.
4. CB-treated lymphocytes have the highest and most significant upregulation of IL1, but IL10 is elevated but not significantly. However, there is no discussion of IL1
Minor
1, IL10 expression is high in bone tissue, such as in Tibia. Why? There may be an abundance of lymphocytes --isn't that odd?
2. GC-MS analysis showed that butyrate concentration in the cecum was higher in the CB-treated group, but not significantly. It seems that CB treatment was insufficient.
3. The photo in Fig3B is unclear and I have no idea what it shows
Reviewer 2 Report
as requested, I reviewed the manuscript (ID ijms-2069058) "Clostridium butyricum can promote bone development by regulating lymphocyte function in laying hens", by Mengze Song, Xuesong Zhang, Guijuan Hao, Hai Lin, and Shuhong Sun.
The proposed manuscript deals with a study of the effects of Clostridium butyricum (CB) as improving supplementation on bone health and immune-mediated bone homeostasis processes in laying hens. The authors described how CB ameliorates the anatomical processes of bone formation, also describing gene expression markers. Moreover, the increase of immune cell types has been put in relation with the CB supplementation. Finally, an experimental setup of in vitro assays, corroborated the major points of their hypothesis.
The data contained in the manuscript are well described and sufficiently interesting to be exploited by experts in the field to obtain interesting advances.
The manuscript is well conceived with balance among the different sections. Overall, the issues in the Discussion contain all the elements for comprehension of the subject, for understanding the cited results and for finding hints to deepen information.
Reviewer 3 Report
Comments to the Author
There is a great need for new nutritional strategies to improve farm animal health And welfare outcomes under industrial production. As such, the objective of this experiment is worthy of investigation. However, the execution of this objective is lacking and unfortunately, the experimental design did not allow the authors to actually test the hypothesis that the CB supplementation in the diet of laying hens would improve performance. This is because the authors did not include different doses of CB in the diet. As such, they were not able to provide any proof that what levels of CB can be adequate. Moreover, the results are too significant but there is no parameter to prove the mode of action and how the CB could be effective. Furthermore, there is no possible way that another researcher could repeat the experiment based on the information given in the methods section. For example, although it seems that the authors measured cytokines but there is no detailed information.
Please add the ethics number
The abbreviations were carelessly added. All abbreviations have to be defined when first presented. ALP in the abstract. OPG, BMD, TRAP, RANK, … in the manuscript.
Lines 53-54 please add references
Lines 69-71 please add references
This sentence has to be moved to the methodology “We supplemented hens with 1 × 108 CFU CB or a saline control by oral gavage.”
Lines 48, 334, the CB was abbreviated several times.